# Role of HIPEC after Complete Cytoreductive Surgery (CRS) in Peritoneal Recurrence of Platinum-Sensitive Recurrent Ovarian Cancer (OC): The Aim for Standardization at Two Reference Centers for CRS

**DOI:** 10.3390/cancers15020405

**Published:** 2023-01-07

**Authors:** Miklos Acs, Michael Gerken, Vanessa Schmitt, Pompiliu Piso, Alfred Königsrainer, Saher Baransi, Can Yurttas, Sebastian Häusler, Philipp Horvath

**Affiliations:** 1Department of General and Visceral Surgery, Hospital Barmherzige Brüder, 93049 Regensburg, Germany; 2Tumor Center—Institute for Quality Management and Health Services Research, University of Regensburg, 93049 Regensburg, Germany; 3Department of General, Visceral and Transplant Surgery, University Hospital Tübingen, 72074 Tübingen, Germany; 4Department of Gynecology and Gynecological Oncology, Florence Nightingale Hospital, 40489 Düsseldorf, Germany; 5Department of Gynecology and Gynecological Oncology, Hospital Barmherzige Brüder, 93049 Regensburg, Germany

**Keywords:** platinum-sensitive, recurrent ovarian cancer, cytoreductive surgery, HIPEC

## Abstract

**Simple Summary:**

The vast majority of patients with epithelial ovarian carcinoma (OC) will relapse during the natural history of their disease. The role of cytoreductive surgery (CRS) in the treatment of recurrent disease has been emphasized by current studies. Adding hyperthermic intraperitoneal chemotherapy (HIPEC) has been evolved to improve DFS and OS. There are currently two convincing studies of HIPEC after complete cytoreduction in the treatment of primary OC, but there is little homogenous data on the role of HIPEC in platinum-sensitive recurrent ovarian cancer, its ideal compound, and its duration. The aim of this study was to analyze the bicentric experience with CRS + HIPEC in patients with platinum-sensitive recurrent epithelial OC in order to standardize the surgical approach. Thus, multimodal therapy was feasible with acceptable morbidity and mortality. Cisplatin monotherapy as a HIPEC compound and a 90 min HIPEC application proved to be the best option for regional additive treatment.

**Abstract:**

Background: This bicentric study evaluated cytoreductive surgery (CRS) combined with hyperthermic intraperitoneal chemotherapy (HIPEC) for platinum-sensitive recurrent ovarian cancer patients. Methods: The data of 88 patients with the first peritoneal recurrence of platinum-sensitive epithelial ovarian cancer who underwent CRS and HIPEC from a prospective HIPEC registry were retrospectively investigated. Endpoints were feasibility, chemotherapeutic compound, time of exposure, complications, and overall survival. Results: The median follow-up was 4.7 years (95%-CI 4.6–5.5). The median age was 55.8 years (IQR: 50.3–66.2). Eighty-four patients (95.5%) had high-grade serous histology. The median peritoneal cancer index was 12.0 (IQR: 7.0–20.5). Sixty-five patients (73.9%) had complete cytoreduction (CCR 0). Thirty-eight patients (43.2%) received HIPEC for 60 min, and fifty patients (56.8%) for 90 min. Eighteen patients (20.5%) had grade III to IV complications. One patient (1.1%) died perioperatively. The overall median survival was 43.1 months (95%-CI 34.1–52.2), and the 5-year survival rate was 39.7%. Only 90 min HIPEC and cisplatin were associated with survival. Conclusion: In well-selected patients with platinum-sensitive recurrent ovarian cancer, survival may correlate with complete CRS and 90 min cisplatin-based HIPEC. We confirmed the results of primary OC studies; therefore, this combination should be used for further analysis in the recurrent situation.

## 1. Introduction

In the past decade, no other treatment algorithm has been as critically scrutinized as the treatment for peritoneal metastases of recurrent OC, a cancer type that is inseparably linked to the synchronous and metachronous occurrence of peritoneal tumor implants. The state-of-the-art treatment for primary ovarian cancer is macroscopic complete resection followed by adjuvant systemic chemotherapy, mostly with carboplatin and paclitaxel [1]. In Germany, approximately 8000 women are diagnosed with ovarian cancer each year, with 80% presenting with locally advanced stages (Fédération Internationale de Gynécologie et d’Obstétrique (FIGO) stage IIIB-IV).

The vast majority of patients with platin-sensitive ovarian cancer will suffer peritoneal recurrence within 5 years of primary diagnosis. The role of surgery in the recurrent disease setting is heterogenous and not yet completely well defined. Prior to the DESKTOP (Descriptive Evaluation of Preoperative Selection Criteria for Operability in Recurrent Ovarian Cancer) trials, available data were of a retrospective nature, comprising a very heterogenous patient population. The DESKTOP-I trial outlined the benefit of a macroscopic complete resection for recurrent ovarian cancer [2]. Upon these results, the AGO-score was defined, including three variables, predicting macroscopic complete resection (ECOG of 0; ascites less than 500 mL and complete resection at primary surgery). This score was later validated in the DESKTOP-II trial [3]. Recently, the benefit of cytoreductive surgery and systemic chemotherapy over systemic chemotherapy alone in recurrent ovarian cancer was outlined in the DESKTOP-III trial [1]. Patients with a macroscopic complete resection, achieved in 75.5% of patients, benefited the most with a median overall survival of 61.9 months. These results emphasized the need for high-quality surgery in recurrent ovarian cancer in order to maximize patients’ benefits.

The addition of hyperthermic intraperitoneal chemotherapy (HIPEC) to the treatment algorithm for recurrent ovarian cancer has been issued intensively in the last couple of years. There are a variety of published randomized trials documenting the benefit of intraperitoneal (IP) chemotherapy, and despite the overwhelming data suggesting a clear benefit of IP therapy in primary (vanDriel) and recurrent ovarian cancer [4,5], it is not recommended by the German guidelines so far [6]. The aim of this bicentric study was to critically analyze a potential clinical benefit in patients with platin-sensitive peritoneal recurrent epithelial ovarian cancer undergoing hyperthermic IP therapy after CRS.

## 2. Materials and Methods

### 2.1. Patients and Study Design

From 2007 to 2020, patients undergoing CRS with adjuvant HIPEC for recurrent ovarian cancer were retrospectively analyzed. All patients suffered from the first peritoneal recurrence of a high-grade and platinum-sensitive ovarian cancer. All patients were operated on in one of the two contributing centers, which are highly experienced in the treatment of peritoneal surface malignancies (110 and 50 procedures annually in Regensburg and Tübingen, respectively). The information was obtained from a prospectively managed database. Data analysis included the following parameters: age; sex; BMI (body mass index); ASA (American Society of Anesthesiologists classification); initial FIGO stage; operative procedures; HIPEC compound, duration, and way of application (open or closed); peritoneal cancer index (PCI) and completeness of cytoreduction (CC-score); length of hospital stay and need of intensive care unit (ICU stay); in-hospital mortality and morbidity rate including re-operation rate; and date of death, date of subsequent second recurrence, and last date alive.

Preoperative diagnostics consisted of a thorough clinical examination, blood tests, and a computed tomography (CT) scan. CT images were acquired by a 128-slice multi-detector spiral CT. The reconstructed slice thickness was 5 mm without gaps between slices. Irresectability was defined as infiltration of the mesenteric axis, retroperitoneal plane, or the pancreatic head. Eligibility for CRS and HIPEC was assessed by a surgical oncologist, a medical oncologist, a radiologist, and a radio-oncologist, all of whom attended the interdisciplinary oncologic team meeting. All patients underwent complete resection at initial surgery, and all patients had received previous platinum-based chemotherapy at first diagnosis. Since our data collection began before the DESKTOP trial, the AGO (Arbeitsgemeinschaft Gynaekologische Onkologie) score did not find general application [7]. A further inclusion criterion was a platinum-free interval of more than 6 months after the administration of the last cycle of chemotherapy. Adverse events were classified according to the Clavien–Dindo complication score, and major complications were defined as Grade ≥ III [8]. In-hospital perioperative mortality was defined as death within 90 days of surgery. Tumors were classified by histology according to the World Health Organization classification.

### 2.2. Cytoreductive Surgery

After laparotomy through a mid-line incision and complete adhesiolysis, the peritoneal cancer index (PCI) was determined following the criteria described by Jaquet and Sugarbaker [9]. Abdominal regions were categorized as the small bowel, consisting of Sugarbaker’s abdominopelvic regions (SAPR) 9 to 12; the upper abdomen, consisting of SAPR 0 to 3; and the lower abdomen/pelvis, consisting of SAPR 4 to 8. Tumor-involved structures were resected along with peritonectomy procedures described by Sugarbaker [10,11], aiming for complete cytoreduction, CC-0 and CC-1 (CC-0 indicates no visible disease; CC-1 indicates nodules smaller than 0.25 cm; CC-2 indicates nodules over 0.25 cm and less than 5 cm; CC-3 indicates nodules larger than 5 cm).

### 2.3. HIPEC

After complete cytoreduction and fashioning of intestinal anastomoses, HIPEC was administered for 60 to 90 min at 42 °C depending on the HIPEC compound using the open- or closed-abdomen technique. The dosage of HIPEC compounds was as follows: cisplatin 75 mg/m^2^ in the Hospital Barmherzige Brüder Regensburg and 50 mg/m^2^ in the University Hospital Tübingen, doxorubicin (15 mg/m^2^), gemcitabine (1000 mg/m^2^), and mitomycin (30 mg/m^2^). The reason for increasing the treatment duration from 60 to 90 min was the result of the prospective randomized trial in primary ovarian cancer by van Driel [12]. After completion of HIPEC, the abdomen was washed out with 3 L of lactated Ringer solution, and the abdomen was closed.

### 2.4. Statistical Analysis

The distributions of continuous data are presented as mean, range (minimum and maximum), median, and interquartile range (IQR). Categorical data are described using absolute frequencies and relative percentages. Metric variables with normal distribution—verified by the Kologorov–Smirnov test—were analyzed for differences in their means using Student’s *t*-test; otherwise, the Mann–Whitney U test was applied. Independence of categorical variables was analyzed with Pearson’s chi-squared test; in the case of small numbers, Fisher’s exact test was used.

The vital status was derived from clinical reports, death certificates, and registration offices. Recurrences were obtained from clinical reports and the tumor center registry, being defined as locoregional relapse and/or recurrence as distant metastases. Overall survival (OS), cumulative recurrence, and recurrence-free survival (RFS) were analyzed from the date of surgery until the first event. Patients’ OS, cumulative recurrence, and RFS were estimated with the Kaplan–Meier method. Survival differences were tested for statistical significance by the two-sided log-rank test (Mantel–Cox); the level of significance was set to 0.05. The follow-up period and survival times were right-censored using 31 October 2021 as the cut-off date. The mean and median follow-up period in years were estimated by the reverse Kaplan–Meier method, constructed by reversing “censor” and “event”.

To determine the influence of patient, tumor, and therapy characteristics on overall and recurrence-free survival, we performed univariable and multivariable regression analyses using Cox proportional hazard models. In multivariable analyses, the hazard ratios (HR) were adjusted for the covariables age, sex, BMI, ASA, histologic type, initial FIGO stage, PCI, CCR score, length of HIPEC (90 vs. 60 min), and HIPEC substances, whereby categories with small numbers were rationally aggregated. Hazard ratios and corresponding 95% confidence intervals (CI) were estimated and considered statistically significant when the CI excluded 1.0 and a two-sided *p*-value was <0.05.

The multicollinearity of variables was checked in advance. Variables were excluded from the model when the variance influence factor (VIF) proved to be >10 in collinearity diagnostics derived from regression analysis. The proportional hazard assumption was tested by inspecting Kaplan–Meier curves and the log-minus-log plots.

The findings of this survey are presented in strict compliance with the Strengthening the Reporting of Observational Studies in Epidemiology (STROBE) statement: guidelines for reporting observational studies [13]. All analyzes were performed using IBM SPSS Statistics software, Version 28.0 (IBM Corp., Armonk, NY, USA).

## 3. Results

### 3.1. Characteristics of Patients, Tumor, Therapy, and Short-Term Outcome

From 2007 to 2020, 88 patients underwent CRS for first recurrence of epithelial ovarian cancer. The median age of the patients was 55.8 years (IQR 50.3–66.2, mean 57.4, range 28.3–77.2), and the median BMI was 24.9 (IQR 22.6–28.4). Over two-thirds (69.3%) showed an initial FIGO stage III, and one patient had a completely resected stage IV tumor. The peritoneal cancer index (PCI) was 12.0 (IQR 7.0–20.5). High-grade serous carcinoma was the most common histological subtype (95.5%), while mucinous carcinoma was diagnosed in two patients (2.3%), and transitional cell carcinoma and unspecified carcinoma each occurred in one patient (1.1%). The patient and tumor characteristics are listed in Table 1.

Eight patients (9%) received previous bevacizumab as induction therapy. Three patients (3.4%) had already been treated with CRS and HIPEC as part of the primary treatment. The median time to first recurrence after primary ovarian cancer diagnosis was 2.3 years (95%-CI 2.0–2.6).

Complete (CC-0) cytoreduction was achieved in 65 patients (73.9%), and in 17 patients (19.3%), a CC-1-status was achieved. In the remaining six patients (6.8%), a debulking situation (CC2-3) was achieved after CRS + HIPEC.

A total of 36 (40.9%) and 33 patients (37.57%) received open and closed HIPEC, respectively, whereas 19 patients (21.6%) underwent open–closed HIPEC application. Regarding the duration of HIPEC, in 38 patients (43.2%), it was performed for 60 min, and in 50 patients (56.8%) for 90 min. The median dosage of cisplatin mono in 47 patients was 85.0 mg (IQR 73.0–95.0%, mean 80.2, range 50.0–107.5). When combined with doxorubicin (35 patients), the median cisplatin dose was higher with 126.0 mg (IQR 90.0–133.6, mean 118.1, range 80.0–157.5). The median dose of doxorubicin here was 26.0 mg (IQR 25.2–28.6, mean 26.5, range 22.5–31.5). The distribution of therapy and surgical procedures is listed in Table 2.

The median hospital stay was 15.0 days (range IQR 13–21). The major morbidity (Clavien–Dindo Grade ≥ III) rate was 21.6% (19 patients). A total of 16 patients (18.2%) had grade III complications, while 2 patients (2.3%) had grade IV complications. Twelve patients (13.6%) required reoperation in the early postoperative period (five surgery site infection, two fascial dehiscence, one intraabdominal bleeding, two anastomotic insufficiency, one small bowel leakage, one bile leakage). One patient (1.1%) died within 90 days postoperatively due to an acute fulminant pulmonary embolism. The short-term outcome variables are listed in Table 3.

### 3.2. Long-Term Outcome

The median follow-up was 4.7 years (95%-CI 2.5–6.9), and mean follow-up was 5.8 years (95%-KI 4.7–7.0). In the complete cohort of 88 patients, 46 (52.2%) died within the follow-up period after surgery of the first recurrence. Median overall survival was 3.6 years (95%-CI 2.8–4.3, 43.1 months), and the 5-year overall survival rate was 39.7% (survival rate after 1 year: 91.1%, 2 years: 79.4%, 3 years: 61.4%) (Figure 1). Figure 2 provides a flow chart depicting the main results.

In univariate analyses of overall survival by the Kaplan–Meier method and Cox regression, patient and tumor characteristics (age, BMI, ASA classification, and histological type) proved to be significant prognostic factors for overall survival. Patients aged 60–69 exhibited a significantly superior survival compared to other age groups, possibly explained by the above-average rate of FIGO stage I patients. Histology showed a significantly inferior result for patients with other than high-grade serous carcinoma. The initial FIGO stage did not show any significance according to overall survival (Figure 3, Table 4).

In univariate analyses of overall survival according to the treatment factors, 90 min HIPEC (HR 0.541, 95%-CI 0.300–0.978; *p* = 0.042) and treatment with cisplatin only (as opposed to cisplatin and doxorubicin, which yielded an HR of 2.269 95%-CI 1.230–4.186; *p* = 0.009) appeared to be significantly superior in terms of overall survival, whereas only a small trend was observed for CCR with a benefit of CC0 resected patients (Figure 4, Table 4).

Age at surgery and HIPEC drugs used were significant independent variables in the multivariate Cox regression analysis for overall survival at level 0.05 (HR cisplatin and doxorubicin vs. cisplatin: 5.788 95%-CI 1.068–31.380; *p* = 0.042; HR age 60–69 vs. age < 50: 0.332, 95%-CI 0.122–0.901; *p* = 0.030) (Table 4).

In the complete cohort of 88 patients, 27 (30.7%) developed a second recurrence within the follow-up period after surgery. The five-year cumulative recurrence rate was 42.0% (recurrence rate after 1 year: 16.6%, 2 years: 33.8%, 3 years: 35.9%).

Sixty-one patients were deceased or had a second recurrence within the follow-up period after surgery of the first recurrence. Median recurrence-free survival was 2.0 years (95%-CI 1.2–2.7, 23.7 months), and the 5-year recurrence-free survival rate was 18.8% (survival rate after 1 year: 76.2%, 2 years: 49.5%, 3 years: 36.0%). In the multivariate Cox regression analysis for recurrence-free survival, only age proved to be an independent prognostic factor.

## 4. Discussion

In this bicentric study, we evaluated the outcome and long-term survival of 88 platinum-sensitive recurrent OC cases who experienced their first relapse and underwent secondary CRS and HIPEC.

A predictive marker for recurrence in epithelial ovarian cancer has not been prospectively verified yet, and only a proportion of patients are eligible for secondary CRS.

The key objective of repeated surgery in recurrent ovarian, fallopian tube, and peritoneal cancer is to resect all visible disease to accomplish complete gross (CCR0) resection, which has been proven to prolong survival [1,14,15]. Selecting the candidates and predicting the likelihood of complete secondary cytoreduction is a complex and elaborate assignment. To achieve this, prognostic models have been developed recently. The iMODEL [16] includes the initial FIGO stage (I/II versus III/IV), complete or incomplete cytoreduction in the primary setting, performance status, progression-free interval after primary treatment (≥16 versus <16 months), CA-125 level at recurrence, and presence of ascites. More widely used is the AGO score [7]. The predictive value of the latter was evaluated prospectively in the DESKTOP II trial [3], where a complete cytoreduction rate of 76% was achieved in appropriately selected patients with a positive AGO score after a platinum-free interval of more than 6 months. In our clinical practice and in this study, in addition to the clinical parameters mentioned above, the main selection criterion to evaluate resectability was a chest/abdominal/pelvic computed tomography (CT). Furthermore, a very useful selection tool the diagnostic laparoscopy, which we perform regularly in our centers to assess the feasibility of cytoreduction prior surgery [17,18]. Interestingly, laparoscopy is not part of these score systems; however, it serves as a proven selection tool [17]. Moreover, it should be addressed that a possible limitation for laparoscopy may be adhesions due to previous surgery. Due to the high number of referrals to our tertiary centers, the presence of ascites and level of CA-125 were of secondary importance if the other predictive values did not exclude multimodal therapy. With these assessments, we could achieve a 74% complete cytoreduction, which is comparable to the above-mentioned and current studies [1,19,20].

### 4.1. Role of Surgery in Recurrent Epithelial Ovarian Cancer without HIPEC

Several retrospective studies have indicated that complete CRS to no visible disease was associated with increased overall survival (OS), compared with leaving any visible residual disease after secondary surgery [15,21]. Furthermore, the superiority of cytoreduction compared with chemotherapy alone in platinum-sensitive recurrent ovarian cancer has also been demonstrated by retrospective studies [15,22]. However, to date, three phase III trials have addressed the role of secondary CRS in comparison with chemotherapy alone, with discordant results. First, the GOG (Gynecologic Oncology Group) 213 trial [19] after randomizing of 485 patients showed 50.6 versus 64.7 months OS after surgery versus chemotherapy alone, respectively, thus showing a missing impact on OS for secondary cytoreductive surgery. Second, the German DESKTOP III trial [1] randomizing 407 patients with platinum-sensitive recurrent ovarian cancer patients with a positive AGO score recently highlighted a significant increase in overall survival after surgery in comparison with the non-surgery group (median 54 versus 46 months, respectively; HR 0.75; 95%-CI 0.59 to 0.96; *p* = 0.02). Third, the Shanghai Gynecologic Oncology Group’s SOC-1 trial [20] randomized 357 patients predicted to be resectable by iMODEL. After 36 months, median follow-up at the interim analysis showed a median OS of 58 versus 54 months for the surgery group versus the chemotherapy alone group, respectively (HR 0.82, 95%-CI 0.57–1.19).

Nevertheless, the long-term survival data were immature at the time of the interim analysis, thus leaving the final results open. It has to be underlined that there was a significant difference in the use of bevacizumab in these studies, and these results have to be interpreted with caution. Yet beyond that, what explains the difference in results in studies where the focus is on surgery itself? In synopsis of these studies, they lack in surgical variables, quantitative measurement of the tumor burden, and accurate assessment of the surgeries, which are provided by PCI and completeness of cytoreductive surgery (CCR) score [9,23], thus making the most crucial part of the treatment and core of the studies incomparable. To further pose this difference, hereby, we report a severely tumor-burdened patient population in whom extensive multivisceral resection was performed with 22.7% (*n* = 20) diaphragmatic resection, 4.5% (*n* = 4) pancreatic tail, 12.5% liver, and 6.8% (*n* = 6) gastric resection with a median PCI of 12. How can this be compared to the high-ranking studies where mainly the site of relapse is indicated but not the extent of resection and surgical variables? Is it conceivable that patients with different tumor dimensions were included and that makes the difference in long-term outcome? This remains conjecture in the absence of comparable quantitative measurements. These observations are shared by other authors as well [24]. Consequently, patients with recurrent ovarian cancer who have even extensive peritoneal carcinomatosis could be included and operated on if a macroscopically complete tumor resection is expected.

There is already proven evidence that quality of cytoreduction [14] and a multidisciplinary approach in the upper abdomen [25] impact survival [24]. Likewise peritoneal cancer index is a proven predictor of survival in primary advanced-stage serous epithelial ovarian cancer [26], and in the present study, we report that PCI, although not significantly, also influences survival in recurrent ovarian cancer PCI ≤ 9 vs. PCI 10–19 (*p* = 0.307).

### 4.2. HIPEC for Epithelial Ovarian Cancer

There is a growing interest in the use of HIPEC for EOC. The aim of this complementary approach is to eradicate any remaining microscopic residual tumor cells after complete cytoreductive surgery with higher intraperitoneal concentrations of cytotoxic chemotherapy with the potentiated and synergistic effect of heat [27]. This additional modality of adding hyperthermic intraperitoneal chemotherapy to secondary cytoreduction is not considered beneficial for the treatment of recurrent EOC, even though OS, PFS, and morbidity data clearly indicate its significance [4,5]. Present and past results dealing with this special issue so far are partly encouraging and, on the other hand, partly contradictory. The first randomized clinical trial was published by Spiliotis et al. [28] in 2015, showing a significant benefit in mean OS for CRS + HIPEC followed by systematic chemotherapy versus CRS with systematic chemotherapy alone (26.7 versus 13.4 months, respectively; *p* < 0.006). Despite many criticizing methodological errors [29], including the inclusion of platinum resistant patients, the HIPEC group’s survival advantage was clearly demonstrated. Our one-arm (CRS + HIPEC) evaluation revealed a 43.1-month overall survival rate and a 39.7% 5-year survival rate, wherein, in contrast to the latter study, patients with initial FIGO Stage I-IIIB were also included. However, the initial tumor stage had no influence on survival in our study when the patients were primarily completely cytoreduced. Similarly, Kim et al. [30] demonstrated a significantly improved OS in their meta-analysis in patients with recurrent EOC (*n* = 491) if they received HIPEC after CRS compared to those without HIPEC after cytoreductive surgery (HR, 0.566; 95%-CI, 0.379–0.844).

On the other hand, in a randomized phase II trial by Zivanovic et al. [31], 98 patients were randomly assigned to carboplatin HIPEC (800 mg/m^2^ for 90 min) or no HIPEC, followed by five or six cycles of postoperative IV carboplatin-based chemotherapy. There was no significant difference in median overall survival (52.5 vs. 59.7 months, respectively; hazard ratio, 1.39; 95%-CI, 0.73 to 2.67; *p* = 0.31).

The largest retrospective multicentric study for recurrent EOC evaluating HIPEC was performed by Bakrin et al. [32], who involved 184 patients out of 246 with platinum-sensitive recurrent EOC. Multimodal treatment resulted in optimal cytoreductive surgery in 92.2% of the total cohort and a median overall survival of 52 months for platinum-sensitive patients. All authors agreed that complete macroscopic cytoreduction is the major prognostic factor for OS [1,28,31].

The milestone trial on HIPEC in primary epithelial ovarian cancer was undoubtedly conducted by the Dutch gynecologic oncologists van Driel and associates [12]. Thereupon, we show a higher CC-0 rate despite the recurrence situation with 73.9% in comparison of 67% and 69% in surgery and surgery plus the HIPEC arm, respectively. Emphasizing that since complete macroscopic cytoreduction has been shown to be the utmost important aspect of survival, we urge joint patient care with oncological surgeons forming a bidisciplinary ovarian team to achieve the best possible operative outcome, on the basis of the Memorial Sloan Kettering Cancer Center model [25]. Our data support this result where the multidisciplinary approach is commonly performed.

This approach could also contribute to reduce the proportion of ileo- or colostomy as a factor decreasing the quality of life (10.3% versus 17%, respectively).

The main arguments of some gynecologic oncologists against intraperitoneal chemotherapy are the increased morbidity caused by elevated toxicity and the consecutive postponement of adjuvant chemotherapy [33,34]. When scrutinizing the morbidity in terms of HIPEC, however, we found no difference between the additional approach and without it. We evaluated a major complication rate (Clavien–Dindo III and IV) of 20.5% and a mortality rate of 1.1% (*n* = 1), also according to the nationwide German HIPEC Registry. The registry’s investigation of 2149 consecutive patients who underwent CRS and HIPEC revealed a 19.3% major complication rate and a 30-day postoperative hospital mortality of 2.3% [35]. Similarly, a recent meta-analysis found a major complication rate of 14% after primary and 15% after recurrent CRS alone (*p* = 0.83). Furthermore, the mortality rates were 1.0 and 0.9%, respectively [36]. Consequently, this equal morbidity rate is rather the result of extensive cytoreduction than of intraperitoneal chemotherapy. However, the above favorable results should be supplemented by two important aspects. In our clinics, the routine administration of the metal binder sodium thiosulfate in the case of cisplatin intraperitoneal chemotherapy has been firmly integrated from 2021, after several researchers have demonstrated the protective effect and reduction of the incidence of acute kidney injury (AKI) [37,38]. In addition, the goal-directed fluid therapy has an indispensable component to minimize toxicity in intra- and early postoperative management, as it was also included in the recent enhanced recovery after surgery (ERAS) protocol [39,40].

Adjuvant chemotherapy is an important adjunct to surgery in the treatment of platinum-sensitive ovarian cancer patients. To maximize the efficacy of the multimodal therapy concept, the timing of the start of this therapy should also be scheduled as early as possible in cytoreduced patients. The fact that in our cohort the operated patients were hospitalized for a median of 15 days (IQR 13–21) makes it ideal for a timely start of chemotherapy. In our interdisciplinary tumor conferences, platinum chemotherapy with maintenance therapy using bevacizumab or PARP inhibitor (poly-((ADP-ribose))-polymerase), depending on the BRCA (breast cancer tumor suppressor gene) status, was recommended for these patients according to the German national guideline [6]. Nonetheless, the vast majority of patients were treated by referral to our tertiary centers, and therefore they received chemotherapy close to their hometowns throughout Germany. This means that we could only obtain sparse data from the tumor centers about the treatments given, and therefore we cannot report in detail on them.

Currently, there is no consensus on the HIPEC protocol to be used. There is a wide variability of HIPEC drugs and their dosage, combination, duration, open versus closed technique, and temperature applied worldwide [41]. However, there is growing evidence that prolonged HIPEC application with cisplatin leads to longer survival. This is shown by the intra-peritoneal exposure of 90 min to cisplatin in both primary ovarian carcinoma [12,42] and in the present cohort of homogeneous recurrent ovarian cancer patients with OS benefit (*p* = 0.039), which should be highlighted. After demonstrating that 60 versus 90 min of HIPEC treatment does not lead to increased postoperative morbidity, that prolonged treatment may improve survival, and that 90 min of HIPEC with cisplatin monotherapy showed superiority over combination therapy, these HIPEC applications could be considered the best to treat ovarian carcinomas. Nevertheless, almost all patients with HIPEC duration of 90 min received cisplatin monotherapy. The high correlation between 90 min HIPEC and cisplatin monotherapy could explain the beneficial effect besides the different doses of cisplatin administered at the two centers (50 mg/m^2^ and 75 mg/m^2^), underscoring the importance of the duration of the treatment. Our study’s results raise an obvious possibility and provide a space of discussion of survival benefit to further increasing the dosage and duration of HIPEC exposure, analogous to the Dutch Phase III trial [12]. The latter requires further pathophysiological and economic considerations.

The French randomized control trial CHIPOR (NCT01376752) is ongoing with an estimated completion date of 2024. Hereby, the OS in platinum-sensitive recurrent ovarian cancer after randomization in CRS only or CRS plus HIPEC will be compared.

### 4.3. Limitations of the Study

Our study has several limitations, such as the nature of a bicentric institutional retrospective analysis and the treatment variability between the two institutions. Moreover, as a retrospective descriptive study, our present research lacked comparison of a control group without HIPEC or chemotherapy alone for a statistical analysis of the effectiveness of CRS/HIPEC and prognostic factors. Furthermore, the sparse data collection in terms of additive chemotherapy reduces the value of the results.

## 5. Conclusions

In our bicentric retrospective study in the treatment of platinum-sensitive ovarian cancer patients by CRS and HIPEC, we highlighted the effectiveness and feasibility of complete cytoreduction by multidisciplinary participation despite extensive tumor involvement. In addition, we demonstrated the superiority of prolonged HIPEC duration with 90 min and cisplatin monotherapy over 60 min and combination treatment.

## Figures and Tables

**Figure 1 cancers-15-00405-f001:**
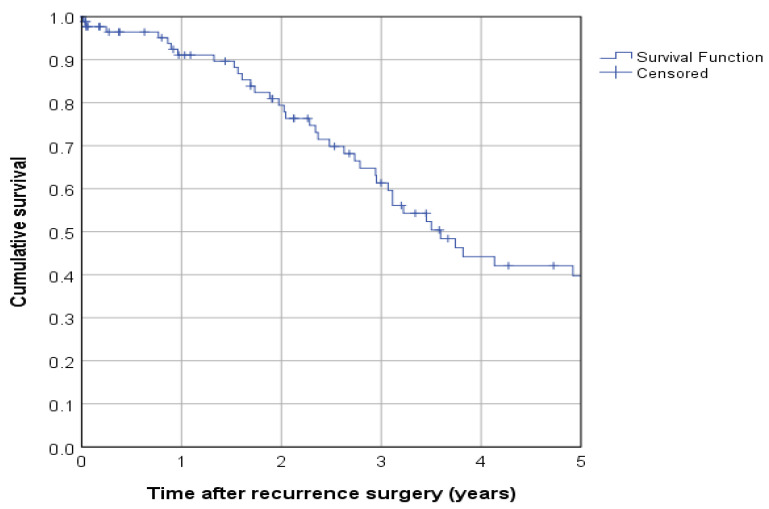
Overall survival in the complete cohort.

**Figure 2 cancers-15-00405-f002:**
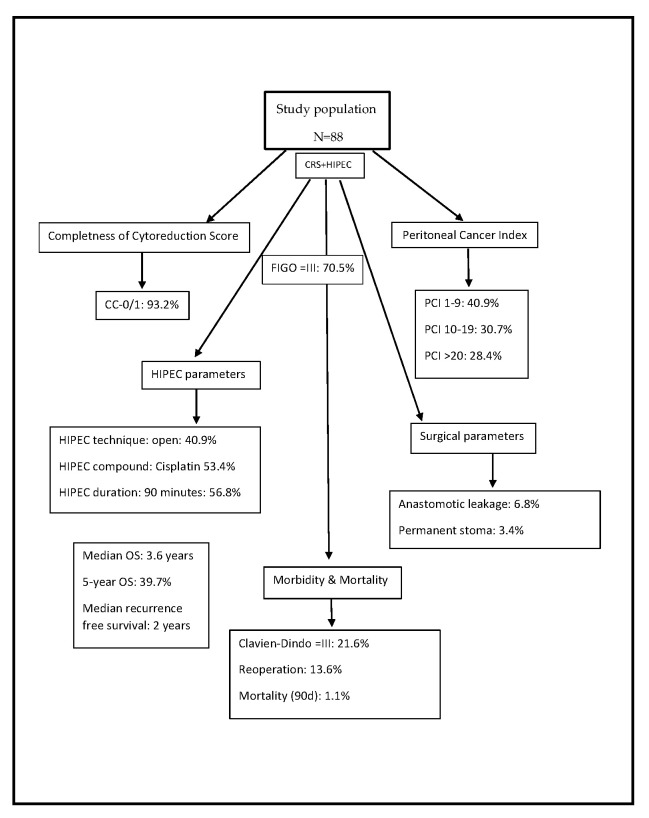
Flow chart depicting the main results.

**Figure 3 cancers-15-00405-f003:**
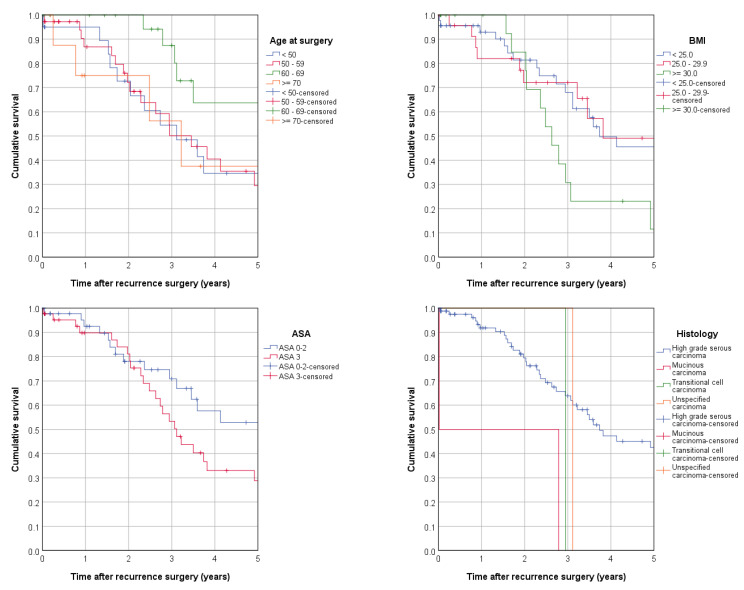
Overall survival according to patient and tumor characteristics (Kaplan–Meier): age at surgery, BMI, ASA classification, histological type, initial FIGO stage, and PCI.

**Figure 4 cancers-15-00405-f004:**
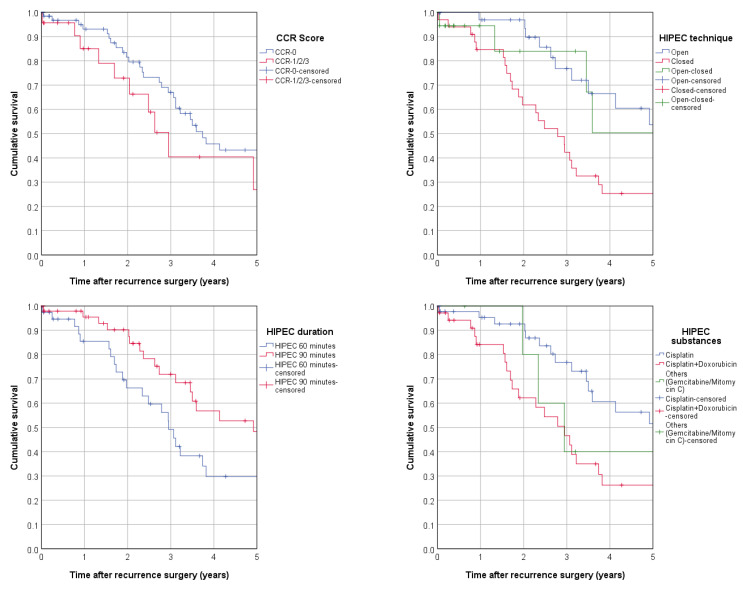
Overall survival according to therapy (Kaplan–Meier): CCR score, HIPEC technique, HIPEC duration, and HIPEC substances applied.

**Table 1 cancers-15-00405-t001:** Patient and tumor characteristics.

		CRS/HIPEC	
		*n*	%
Age at surgery	<50	21	23.9%
	50–59	36	40.9%
	60–69	22	25.0%
	≥70	9	10.2%
Age	Mean/range	57.4	28.3–77.2
	Median/IQR	55.8	50.3–66.2
Time to first recurrence (years)	Mean/95%-CI	3.4	2.7–4.0
	Median/95%-CI	2.3	2.0–2.6
BMI	<25.0	45	51.1%
	25.0–29.9	25	28.4%
	≥30.0	18	20.5%
BMI	Mean/range	25.9	16.3–41.8
	Median/IQR	24.9	22.6–28.4
ASA	ASA 0–2	46	52.3%
	ASA 3	42	47.7%
ASA	Mean/range	2	0–3
	Median/IQR	2	2–3
Histology	High-grade serous carcinoma	84	95.5%
	Mucinous carcinoma	2	2.3%
	Transitional cell carcinoma	1	1.1%
	Unspecified carcinoma	1	1.1%
Initial FIGO stage	I	12	13.6%
	II	14	15.9%
	IIIA IIIB IIIC	2 18 41	2.3% 20.5% 46.6%
	IV	1	1.1%
PCI	1–9	36	40.9%
	10–19	27	30.7%
	≥20	25	28.4%
PCI	Mean/range	14.0	3.0–39.0
	Median/IQR	12.0	7.0–20.5
	Total	88	100.0%

**Table 2 cancers-15-00405-t002:** Distribution of therapy and surgical procedures.

		CRS/HIPEC	
		*n*	%
CCR score	CCR-0	65	73.9%
	CCR-1	17	19.3%
	CCR-2/3	6	6.8%
Length of surgery (min, without HIPEC)	Mean/Range	439	113–885
	Median/IQR	404	310–565
HIPEC technique	Open	36	40.9%
	Closed	33	37.5%
	Open-closed	19	21.6%
HIPEC duration	HIPEC 60 min	38	43.2%
	HIPEC 90 min	50	56.8%
HIPEC substances	Cisplatin	47	53.4%
	Cisplatin + Doxorubicin	35	39.8%
	Gemcitabine	3	3.4%
	Mitomycin C	3	3.4%
Parietal peritonectomy	Yes	77	87.5%
Peritonectomy pelvis	Yes	59	67.0%
Peritonectomy omental bursa	Yes	16	18.2%
Peritonectomy right upper quadrant	Yes	48	54.5%
Peritonectomy left upper quadrant	Yes	33	37.9%
Thoracic drainage	Yes	9	10.2%
Diaphragm resection	Yes	20	22.7%
Hepatic capsule resection	Yes	10	11.4%
Appendectomy	Yes	7	8.0%
Colon resection	Yes	36	40.9%
Small bowel resection	Yes	25	28.4%
Low anterior rectum resection	Yes	29	33.0%
Splenectomy	Yes	29	33.0%
Pancreatectomy (tail)	Yes	4	4.5%
Cholecystectomy	Yes	27	30.7%
Greater omentectomy	Yes	33	37.5%
Lesser omentectomy	Yes	13	14.8%
Liver resection	Yes	11	12.5%
Stomach resection	Yes	6	6.8%
Hysterectomy	Yes	2	2.3%
Adnexectomy	Yes	1	1.1%
Resection other organs	Yes	10	11.4%
Anastomosis small bowel–small bowel	Yes	13	14.8%
Anastomosis stomach–small bowel	Yes	1	1.1%
Anastomosis small bowel–colon	Yes	26	29.5%
Anastomosis colon–colon	Yes	5	5.7%
Anastomosis colon–rectum	Yes	29	33.0%
Anastomosis small bowel–rectum	Yes	4	4.5%
Colostomy	Yes	2	2.3%
Ileostomy	Yes	7	8.0%
Permanent colostomy	Yes	1	1.1%
Permanent ileostomy	Yes	2	2.3%
	Total	88	100.0%

**Table 3 cancers-15-00405-t003:** Distribution of short-term outcome variables.

		CRS/HIPEC	
		*n*	%
Length of hospital stay	Mean/range	20.4	8–93
	Median/IQR	15.0	13–21
Length of ICU stay	Mean/range	3.8	1–15
	Median/IQR	3.0	2–4
Erythrocyte concentrates	Yes	19	21.6%
Fresh frozen plasma	Yes	36	40.9%
Human albumin	Yes	3	3.4%
Chest drain	Yes	6	6.8%
Pleural puncture	Yes	5	5.7%
Complication grade (Clavien–Dindo)	0	44	50.0%
	II	25	28.4%
	III	16	18.2%
	IV	2	2.3%
	V	1	1.1%
Complication grade (Clavien–Dindo)	Mean/Range	1	0–5
	Median/IQR	1	0–2
Anastomotic insufficiency	Yes	6	6.8%
Pneumonia	Yes	4	4.5%
Pulmonary embolism	Yes	13	14.8%
Pleural effusion	Yes	9	10.2%
Urinary tract infection	Yes	6	6.8%
Renal insufficiency	Yes	1	1.1%
Pancreatitis	Yes	2	2.3%
Pancreatic fistula	Yes	2	2.3%
Deep vein thrombosis	Yes	4	4.5%
Surgical site infection	Yes	9	10.2%
Mortality (90 d)	Yes	1	1.1%
Reoperation	Yes	12	13.6%
	Total	88	100.0%

**Table 4 cancers-15-00405-t004:** Overall survival according to therapy, patient, and tumor characteristics (Cox regression).

Variable	Category	Univariable Regression				Multivariable Regression			
		HR *	Lower 95%-CI	Upper 95%-CI	*p*	HR *	Lower 95%-CI	Upper 95%-CI	*p*
CCR score	CCR-0	1.000				1.000			
	CCR-1/2/3	1.289	0.631	2.633	0.486	0.718	0.238	2.166	0.557
HIPEC duration	HIPEC 60 min	1.000				1.000			
	HIPEC 90 min	0.541	0.300	0.978	0.042	3.048	0.569	16.343	0.193
HIPEC substances ^&^	Cisplatin	1.000			0.032				0.114
	Cisplatin + doxorubicin	2.269	1.230	4.186	0.009	5.788	1.068	31.380	0.042 ^$^
	Gemcitab./mytomycin	1.687	0.493	5.774	0.405	4.385	0.411	46.806	0.221
Age at surgery	<50	1.000			0.098	1.000			0.081
	50–59	1.154	0.565	2.356	0.694	1.272	0.523	3.096	0.596
	60–69	0.444	0.180	1.094	0.078	0.332	0.122	0.901	0.030 ^$^
	≥70	1.497	0.523	4.284	0.452	1.137	0.318	4.068	0.844
BMI	<25.0	1.000			0.095	1.000			0.421
	25.0–29.9	1.076	0.530	2.187	0.839	1.250	0.519	3.011	0.619
	≥30.0	2.155	1.049	4.428	0.037	1.876	0.734	4.793	0.189
ASA	ASA 0–2	1.000				1.000			
	ASA 3	1.945	1.048	3.610	0.035	1.251	0.545	2.871	0.597
Histology	High-grade serous Ca	1.000				1.000			
	Others	2.959	1.025	8.540	0.045	2.555	0.558	11.714	0.227
Initial FIGO stage	I	1.000			0.409	1.000			0.796
	II	1.624	0.523	5.038	0.401	0.698	0.166	2.937	0.624
	III/IV	1.783	0.689	4.610	0.233	1.019	0.302	3.443	0.975
PCI	0–9	1.000			0.528	1.000			0.382
	10–19	0.689	0.328	1.451	0.327	1.087	0.407	2.905	0.868
	≥20	1.092	0.531	2.244	0.812	1.951	0.697	5.461	0.203

* HR: hazard ratio, CI: confidence interval. ^$^
*p*-value in the row of reference category denotes significance of complete variable. ^&^ due to collinearity with HIPEC technique, HIPEC substances were excluded from the multivariable analysis.

## Data Availability

The data presented in this study are available on request from the corresponding author.

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
