# Peer review of "Role of HIPEC after Complete Cytoreductive Surgery (CRS) in Peritoneal Recurrence of Platinum-Sensitive Recurrent Ovarian Cancer (OC): The Aim for Standardization at Two Reference Centers for CRS"

_cancers, 2023, doi:10.3390/cancers15020405_

Round 1

Reviewer 1 Report

Very interesting paper, though the results can be applied in specific selected cases

Author Response

Thank you very  much for reviewing our manuscript.

Reviewer 2 Report

The very topic seems to of the publication seems to very be interesting. The authors present a publication in which they analyse the patients with platinum sensitive recurrent epithelial ovarian cancer with the focus to standardize the surgical approach. This is a very important topic. Authors have chosen very representative group of patients to this study. The manuscript is written with a clear and understandable English. However, I have a few concerns:

1.    There are some stylistic errors that should be corrected.

2.    The graphical presentation of the results is not very clear and therefore difficult to interpret.

3.    The introduction should be supplemented with information on ovarian cancer which will provide sufficient background.

Author Response

#2

The very topic seems to of the publication seems to very be interesting. The authors present a publication in which they analyse the patients with platinum sensitive recurrent epithelial ovarian cancer with the focus to standardize the surgical approach. This is a very important topic. Authors have chosen very representative group of patients to this study. The manuscript is written with a clear and understandable English. However, I have a few concerns:

  1. There are some stylistic errors that should be corrected.

The manuscript has been revised and changed paragraphs are marked in purple.

  1. The graphical presentation of the results is not very clear and therefore difficult to interpret.

Thank you for this comment. We believe that the tables und Kaplan-Meier curves are quite comprehensible and readable, therefore we decided not to change the layout. The tables and the Kaplan-Meier curves are large in size and Table 1 to 3 lists all treatment- and patient-relevant factors in n(number) and percentage (%) of the total study population. Table 4 sums ups the statistical results. If the reviewer indicates which changes are deemed necessary, we will provide revised graphics but otherwise it is difficult for us to change them, because no clear suggestion for improvement was provided.

  1. The introduction should be supplemented with information on ovarian cancer which will provide sufficient background.

Thank you for this comment. Being honest we did not quite get the point of the reviewers’ comment. The introduction contains the main information on IP therapy in (recurrent) OC including the DESKTOP-trials and the Van driel trial. We believe that the introduction gives the reader a compact and informative overview of the current and debatable issue of IP therapy in OC. If the reviewer can indicate which literature is lacking, we will of course add these manuscripts but for the time being please do not be disappointed if we leave the introduction unchanged.

Reviewer 3 Report

The authors present their efforts in identifying an ideal approach to the radical treatment of recurring platinum-sensitive ovarian cancer with peritoneal metastases by means of cytoreductive surgery and HIPEC. Ovarian cancer patients are commonly confronted with recurrences, even if a primary complete excision was possible. This study enriches the current knowledge  by highlighting the possibility of increasing survival through cytoreduction and HIPEC. However there are several issues that must be addressed:

- the text must be thoroughly revised, several paragraphs used during for formatting of the text were not deleted (at the beginning of the Results and Discussion sections); also the text should be revised for fluency, some of the sentences are hard to follow or unclear;

- although this limitation is mentioned by the authors, one concern is regarding the absence of a control group; selection of patients with similar characteristics to those of the study group, that underwent cytoreductive surgery and systemic chemotherapy, would allow for a more realistic comparison of the potential benefits of HIPEC over systemic chemotherapy regimens; it is difficult to support the recommendations expressed in the conclusion of the abstract;

- regarding the multivariate analysis could the authors explain why were variables that were found not significant in the univariate analysis, included in the multivariate analysis?

- in table 4 the p column for the multivariate analysis is missing.

Author Response

#3

The authors present their efforts in identifying an ideal approach to the radical treatment of recurring platinum-sensitive ovarian cancer with peritoneal metastases by means of cytoreductive surgery and HIPEC. Ovarian cancer patients are commonly confronted with recurrences, even if a primary complete excision was possible. This study enriches the current knowledge  by highlighting the possibility of increasing survival through cytoreduction and HIPEC. However there are several issues that must be addressed:

- the text must be thoroughly revised, several paragraphs used during for formatting of the text were not deleted (at the beginning of the Results and Discussion sections); also the text should be revised for fluency, some of the sentences are hard to follow or unclear;

Thank you for your comment. Now the whole text was checked and revised  for fluency. These changes have been corrected everywhere in a traceable manner, for better controllability.

- although this limitation is mentioned by the authors, one concern is regarding the absence of a control group; selection of patients with similar characteristics to those of the study group, that underwent cytoreductive surgery and systemic chemotherapy, would allow for a more realistic comparison of the potential benefits of HIPEC over systemic chemotherapy regimens; it is difficult to support the recommendations expressed in the conclusion of the abstract;

Thank you for your comment. Now the consclusion was completed with may in the absence of of a control group with systemic chemotherapy.

- regarding the multivariate analysis could the authors explain why were variables that were found not significant in the univariate analysis, included in the multivariate analysis?

Thank you for your comment. We apply the mentioned strategy of preselection of variables on the basis of p-values from univariable analyses when we develop prediction or prognosis models. In the present case the focus of the analysis is on the effect of therapy, i.e. cytoreductive surgery and HIPEC. In this context it is more meaningful to adjust for all potenial confounders, even though they do not prove to be significant in univariable analyses.

- in table 4 the p column for the multivariate analysis is missing.

Thank you for your important hint. We added a column with the missing p-values in table 4.

Round 2

Reviewer 3 Report

The authors have addressed the highlighted study. Although the study is missing a control group, the results are well presented and warrant publication.

Author Response

Thank you for the re-evaluation of our manuscript.